# Parameterization of Biomechanical Variables through Inertial Measurement Units (IMUs) in Occasional Healthy Runners

**DOI:** 10.3390/s24072191

**Published:** 2024-03-29

**Authors:** Álvaro Pareja-Cano, José María Arjona, Brian Caulfield, Antonio Cuesta-Vargas

**Affiliations:** 1Grupo Clinimetría en Fisioterapia (CTS 631), Department of Physiotherapy, Faculty of Health Sciences, University of Málaga, 29071 Málaga, Spain; aparejacano@gmail.com (Á.P.-C.); jmarjonac@gmail.com (J.M.A.); 2Instituto de Investigación Biomédica de Málaga y Plataforma en Nanomedicina (IBIMA Plataforma Bionand) Grupo Clinimetria (F-14), 29590 Málaga, Spain; 3Faculty of Sciences and Technology, University Isabel I, 09003 Burgos, Spain; 4School of Public Health, Physiotherapy and Sports, University College Dublin, D04 C1P1 Dublin, Ireland; b.caulfield@ucd.ie; 5Insight Centre, University College Dublin, D04 N2E5 Dublin, Ireland

**Keywords:** running, inertial measurement units (IMUs), biomechanics, kinematics

## Abstract

Running is one of the most popular sports practiced today and biomechanical variables are fundamental to understanding it. The main objectives of this study are to describe kinetic, kinematic, and spatiotemporal variables measured using four inertial measurement units (IMUs) in runners during treadmill running, investigate the relationships between these variables, and describe differences associated with different data sampling and averaging strategies. A total of 22 healthy recreational runners (M age = 28 ± 5.57 yrs) participated in treadmill measurements, running at their preferred speed (M = 10.1 ± 1.9 km/h) with a set-up of four IMUs placed on tibias and the lumbar area. Raw data was processed and analysed over selections spanning 30 s, 30 steps and 1 step. Very strong positive associations were obtained between the same family variables in all selections. The temporal variables were inversely associated with the step rate variable in the selection of 30 s and 30 steps of data. There were moderate associations between kinetic (forces) and kinematic (displacement) variables. There were no significant differences between the biomechanics variables in any selection. Our results suggest that a 4-IMU set-up, as presented in this study, is a viable approach for parameterization of the biomechanical variables in running, and also that there are no significant differences in the biomechanical variables studied independently, if we select data from 30 s, 30 steps or 1 step for processing and analysis. These results can assist in the methodological aspects of protocol design in future running research.

## 1. Introduction

Running, a physical activity with many health benefits [1], is influenced by intrinsic and extrinsic factors [2]. Intrinsic factors include biomechanical variables, categorised as kinetic, kinematic, spatiotemporal or running descriptors, and neuromuscular factors [3].

Measuring and understanding biomechanical variables is essential in enhancing performance and preventing and treating running-related injuries [4]. The gold standard methods to measure biomechanical variables in running, including force platforms and optoelectronic motion capture systems [5,6], are expensive and limited mainly to the laboratory setting [7]. This has led to the proliferation of the measurement of biomechanical variables with inertial measurement units (IMUs). This approach offers a reliable and accurate method to measure human motion, and it can be used both in the natural environment of running and on the treadmills [7,8,9].

IMUs are typically equipped with triaxial accelerometers, gyroscopes and magnetometers that measure acceleration, orientation and direction in the three axes of space at high frequencies [10,11]. By analysing the patterns of the acceleration and gyroscope signals of the IMUs, initial contacts and toe-offs can be detected [6,12,13,14]. With the detection of these gait events, it is possible to subsequently calculate spatiotemporal variables, such as step rate, ground contact time, flight time, and step time [15,16,17,18,19,20], and kinematic variables, such as the vertical center of mass displacement (COM) [21]. In addition, from a kinematic parameter, such as acceleration, kinetic variables, such as ground reaction forces (GRF), shock attenuation (SA), and the peak positive attenuation of the tibia can be estimated [22]. These kinetic variables are key to explaining the impact forces experienced by the subjects and also their absorption capacity during running [11,18,21,23,24,25].

There are many factors that should be taken into account when considering the use of IMUs to quantify biomechanical variables during running. The range of variables of interest, the number and locations of sensors, and the sampling period needed to acquire an adequate running representation are some of these parameters, and are not specified in any measurement protocol [8]. Given the lack of established protocols, a wide variety of studies apply one protocol or another depending on their preferences, making it difficult to interpret the results. While some studies measure spatiotemporal and kinetic or kinematic variables together in the same protocol [18,23,26], most studies focus on one or a few variables from the same kinetic [27], kinematic [28] or spatiotemporal group [29]. However, most studies include spatiotemporal variables, as the step rate or step frequency variable is one of the reference variables used to describe the running dynamics. This variable is one of the most commonly studied variables when measuring running with IMUs, such as ground contact time, stride or step length, and tibial acceleration [8].

Data from IMUs placed on different body locations can be processed to derive the same variables [11,30,31], though the results obtained from different locations may differ. This should be considered when comparing results from different studies. For other kinetic variables, such as SA, it is essential to place the sensors at points adjacent to the area we want to measure [32,33,34]. Thus, to calculate the SA of the lumbar area, a sensor must be placed just above and below this vertebral segment [11]. Many studies rely on a sensor usually placed on the lower limbs [8]. However, this limits the ability to extract other relevant variables. Variables such as COM or lumbar SA cannot be obtained in these cases, making it difficult to establish correlations between groups or variables, such as kinetic, kinematic, and spatiotemporal variables.

It is essential to consider the time or steps taken to determine the duration or distance of measurement protocols. Sometimes, different studies with different samples yield similar results. For instance, in a study by Smith et al. [28] the COM variable was measured in runners for 30 s, while in a study by Schütte et al. [35] 20 steps were measured, resulting in almost the same displacement in centimetres. However, studies usually yield different results if they do not have the same sample period, making it difficult to compare them. This sample disparity is evident in the review by Manson et al. [8], where out of 131 articles, 50 selected a certain number of steps or strides. In contrast, 42 selected a period equal to or less than 60 s. In general, previous studies using IMUs to examine running biomechanics have reported on one or two of the three categories of biomechanical variables outlined above, and no study to date has simultaneously quantified all three categories.

This study aims to add value to the field by simultaneously quantifying spatiotemporal, kinematic, and kinetic variables during treadmill running. To obtain this cross-category measurement capability, we have employed a -IMU set-up, one on each shank and two on the lumbar spine. As the effect of the different measurement durations on the running is unknown, we have extracted and compared variables across three different sampling periods. A one-step analysis was conducted to have each subject as a sample or reference; 30 s was selected due to the few studies that selected 30 s, and, in order to homogenize the sample to be analysed, the selection of steps was established in 30 steps.

Therefore, the objectives of this study are to parameterise the biomechanical variables of running through a measurement set-up with four IMUs to extract the kinematic, spatiotemporal variables, and kinetic variables, to describe the relationships between these variables, and to investigate the effect of different data sampling approaches.

## 2. Materials and Methods

### 2.1. Experimental Design

This is a descriptive cross-section laboratory study. All measurements were carried out on the same day, with no follow-up.

### 2.2. Participants

A convenience sampling of 22 healthy occasional runners was selected, of which 18 were male (age: 29 ± 6 years, lower limb length: 92 ± 6 cm, height: 176 ± 7 cm, mass: 77 ± 9 kg, body mass index: 0.22 kg/m^2^ ± 0.02) and 4 were female (age: 26 ± 2 years, lower limb length: 86 ± 3 cm, height: 159 ± 2 cm, mass: 55 ± 4 kg, body mass index: 0.17 kg/m^2^ ± 0.01) (Table 1). They were students from the Faculty, and participated voluntarily. The inclusion criteria were: (i) aged 18–40-years-old, (ii) no pain during running for at least three months, and weekly running distance between 5 and 50 km or 2–3 running sessions per week. A participant who presented (i) concurrent lower limb injuries, (ii) lumbar diseases or deformities, and (iii) the presence of rheumatoid, neurological, or degenerative diseases, was excluded. All participants provided written informed consent, in line with institutional review board procedures.

### 2.3. Experimental Procedure and Data Collection

The testing and running protocol was explained and performed in the Human Movement Laboratory, Faculty of Health Sciences (University of Málaga) (see Appendix A: flow chart of the protocol study). The participants wore shorts and their usual running shoes. Leg length during standing (greater trochanter to ground) was measured using a fabric tape [11,30].

The Shimmer3 IMU (SHIMMER^TM^, Shimmer Research, Dublin, Ireland) (51 × 34 × 14 mm) was utilised to measure accelerations and angular velocities. Before the measurements, each IMU was calibrated and configured to stream triaxial accelerometer (±16 g) and tri-axial gyroscope (±500°/s) data at 512 Hz via Bluetooth to a tabletop in the Shimmer Consensys program. Data were obtained from 4 IMUs attached to the body of the volunteer, tied with an elastic band with Velcro closure, and the inelastic tape was added to the band to prevent secondary movements (Figure 1) [31]. The placement of four IMUs was an adaptation of other set-ups from other authors [11,30,36]. Two sensors were placed in the centre of the medial anterior aspect of each shinbone [37,38]. The centre of the tibia was calculated by measuring its length, from the medial condyle to the medial malleolus, and dividing this length by two. Two other sensors were placed in the lumbar area in the skin overlying the T12/L1 (“mid-back”) and L5/S1 (“low-back”) vertebral levels to isolate shock attenuation within the lumbar vertebral column [11]. The T12-L1 IMU was attached to the skin with double-sided tape and elastic tape on top, to prevent them from remaining in the air due to lumbar lordosis.

The vertical axes (y) of each IMU in the lumbar back were aligned with the craniocaudal axis of the spine, and the transverse (x) axes were aligned with the dorsoventral axis of the volunteers. In the tibiae, the alignment of the accelerometers with the body axes was the same as in the lumbar, but the data from both IMUs were squared first, and the square root was made after, to make them equal in sign.

After placement, participants carried out a warm-up that consisted of some mobility exercises for the lower limbs and lumbar spine, such as rotations and flexion extensions of the joints and fifteen bodyweight squats. The measurement test was a run of 6 min on a non-instrumented treadmill (Salter PT 298-STR, Salter, Barcelona, Spain) at the self-selected pace speed. Participants were instructed to run at a self-selected speed equivalent to the speed they would normally employ for a 10 k run. After accommodating to the treadmill for the first 5 min [39], the sixth minute of running was recorded. The three data sections were selected from that minute: the first 30 s, the first 30 steps, and the first step.

### 2.4. Variables

The running-related variables or spatiotemporal outcomes were preferred speed (PS), the running speed at which participants ran comfortably, in kilometres per hour (km/h); step rate (SR), the number of times the foot contacts with the ground per minute; the ground contact time (GCT), the time from when one of the two feet contacts with the floor to when the foot lifts off in a stride, in milliseconds (ms); the stride time (ST), the sum of the contact time of one foot plus the flight time of the same, in milliseconds (ms); the flight time (FT), the time from toe-off to initial ground contact in a stride for a single limb, in milliseconds (ms); and number of steps of the right foot. The duty factor ratio (DF) was calculated as the ratio of ground contact time to stride time. The kinematic variables were the vertical displacement of the estimation of the whole-body centre of mass (COM), in centimetres (cm), and the sagittal plane knee range of motion (ROM), in degrees. The estimated kinetic variables were the ground reaction force (GRF), in newtons (N); the ground reaction force normalized by the body weight or mass of the participants (GRFbw), in body weights (BW); the peak power spectral density of the region T12-L1, (PPSD) (g^2^/Hz); the peak power spectral density of the region L1-S1 (PPSDL) (g^2^/Hz); the mean power spectral density T12-L1 vertebrae (MPSDM) (g^2^/Hz); the mean power spectral density of L1-S1 vertebrae (MPSDL) (g^2^/Hz); the mean shock attenuation (SA) (dB/Hz), and the peak positive tibial acceleration (PPA) (g).

### 2.5. Data Analysis

Six signals were collected from each IMU: accelerometer x, y, z and gyroscope x, y, z. Raw data were later analysed using a custom-written programme (“Data Plotter”) in MATLAB (R2014b, The MathWorks, Natick, MA, USA). From the raw data of each runner, we selected three windows of data to analyse and to do the statistical analysis: (1) in the first window, completed steps were collected over 30 s; (2) In the second, the first 30 complete steps, 15 with each leg, were recorded; and (3) in the third window, the first complete step of each foot was analysed (Figure 2). Once the time windows and steps were selected, the specific variables were calculated as part of the process. The formulas utilised either accelerometry data, gyroscope data, or a combination of both, depending on the calculated variable. Additionally, one or more IMUs were used to measure various body parts, including the left or right tibia, both tibias, sacrum S1, lumbar L1, or a combination of sacrum and lumbar, depending on the calculated variable. In order to clarify the process, we created a summary table (see Appendix A).

#### 2.5.1. Running Descriptors

We analysed the accelerometer and gyroscope data of the Y axis from the IMU at the shinbones to calculate spatiotemporal variables. We used both left and right tibia IMUs to calculate the step rate variable. We determined foot strikes and toe-offs based on Gantz et al.’s method. To achieve this, we utilised the acceleration data, which was filtered using a Butterworth fourth-order low pass filter with a 2 Hz cutoff frequency. The data was then smoothed through integration and differentiation with the help of continuous wavelet transformation [40]. Finally, we calculated the contact time according to Reenalda et al.’s approach [21].

#### 2.5.2. Kinematic Variables

Concerning the kinematic variables, the whole-body COM variable was extracted from the data of the IMU of the low-back vertebral level. A double integration of the *y*-axis accelerations was performed, and a high-pass Butterworth filter with a cutoff frequency of 0.5 Hz was then applied to eliminate drift [21,41]. For the knee ROM, the tibia’s angle was calculated using a device’s quaternions. To obtain this, multiple sensor data were integrated through a Kalman filter. The process began with data fusion from accelerometers, gyroscopes, and magnetometers, which produced quaternions that represent the spatial orientation of the sensor. These quaternions are crucial in determining the orientation of specific anatomical segments. The gravitational vector defined the longitudinal axis (*z*-axis) of the body segments, while the longitudinal axis of the foot was perpendicular to this gravitational vector. Each quaternion was converted into a rotation matrix, which was then used to calculate the joint angles according to the Cardan angle convention (ZYX sequence in this case) [21]. This method ensured that joint angles, including that of the tibia, were calculated accurately, taking into account the current orientation of the segment relative to a fixed reference frame.

#### 2.5.3. Kinetic Variables

We used the accelerometer data from the IMUs to estimate the kinetic variables. We calculated the root mean square (RMS) and the ground reaction force (GRF) by taking the means of the data in 30-s and 30-step intervals [42,43]. The GRF variable was then normalised by each individual’s body weight (mass) to reduce the variability when comparing [44,45]. The signal was then processed to detect the local maximum in the data [46], and the de-trend process was performed to remove the linear trend. After this, a bandpass filtering between 10 Hz and 60 Hz with a second-order low-pass Butterworth filter was made, according to Lieberman [11], and a spectral analysis was made to calculate the power spectral density (PSD) using the Fast Fourier Transformation [47,48]. Finally, with this, the mean shock attenuation (MSA) and the peak power of the different study areas were obtained [32,34,47], and the PPA data was calculated from the accelerometer of the mid-back level [21].

### 2.6. Statistical Analysis

A descriptive analysis of all the study variables was conducted to find the mean and standard deviation. After this, the Shapiro–Wilk test was performed to check if the variables corresponded to a normal distribution. Inferential statistics were performed to examine the relationship between spatiotemporal, kinematic, and kinetic variables. Correlations were performed using the bivariate analysis of Rho’s Spearman, considering very weak (0–0.19), weak (0.2–0.39), moderate (0.4–0.59), strong (0.6–0.79), and very strong (≥0.8) [49]. In addition, multivariate linear regression models were performed and adjusted by BMI. A one-factor ANOVA analysis was performed, and subsequently, a Bonferroni posthoc test was performed to specify the differences between variables in the three groups of 30 s, 30 steps and 1 step (Appendix A). Statistical analysis was performed using the IBM SPSS Statistics program Version 25 (Armonk, NY, USA), with an alpha level of significance of *p* < 0.05.

## 3. Results

Table 2 displays the results of the spatiotemporal variables, while Table 3 shows the kinetic and kinematic variables.

Correlations between variables were calculated for the different sampling strategies of 30 s, 30 steps and 1 step (Appendix A). Very strong associations were only found between the variables whose formulas share the same unit of measurement (as we summarized in the Appendix A).

No significant associations were found between the forces of the lower limbs (PPA) with the forces of the lumbar area (PSDs), or between the ground reaction forces (GRF and GRFbw) and the attenuation of these forces in the lumbar spine (SA). In the three correlation arrays, very strong, strong and moderate associations were found, being mixed since the direction was positive or inverse, depending on the variable type. The direction of the correlations was observed to be equal in the three selections, except for the association between DF and ST, which in the selections of 30 s and 30 steps was positive (although not significant), and in the selection of 1 step was significant and negative (−0.827).

Based on the correlations observed among different variable families, such as running descriptors, kinetic, and kinematic, we found that the kinematic variable COM had specific associations with spatiotemporal variables when selecting 30 steps. The first association that was observed was a moderate inverse association with SR (−0.488), while the other two associations were moderate but positive with the variables GCT (0.506) and ST (0.516). There were also associations between the kinematic ROM and kinetics variables in the three sample selections. In the 30-s selection, a moderate inverse association existed between ROM and MPSDMID (−0.467) and PPSDMID (−0.501). In the 30-step selection, there was a moderate inverse association between ROM and the MPSDM variable (−0.447), and a positive association with the PPA variable (0.500). When a single step was selected, there was a moderate positive association with GCT (0.469) and an inverse association with the variables PPSDL (−0.425) and PPSDM (−0.470).

Table 4 summarizes the models found in the linear regression analysis. Multiple linear regression analysis in the 30-s selection showed that the FT, DF and GCT variables, after adjusting by BMI, shared 99% of the SR variance; and ST shared 95.2% of SR variance. When 30 steps were selected, all linear regressions were adjusted by BMI. FT, DF and GCT variables shared 93.6% of SR variance; and ST shared 69.1% of the SR variance. The pairwise analysis showed no significant result by comparing variables in 30 s, 30 steps, or 1 step (Appendix A).

## 4. Discussion

This study proposes a 4-IMU data capture and processing approach that can be used to simultaneously describe kinetic, kinematic, and spatiotemporal variables during treadmill running. We also investigated the relationships between these spatiotemporal, kinematic, and kinetic variables, and described differences associated with different data sampling and averaging strategies.

Our results suggest that the 4-IMU approach to data capture and the data processing strategies described in the methods section represent a feasible approach to simultaneously measuring kinetic, kinematic, and spatiotemporal variables during treadmill running. The data capture process was easy to implement, and the method presented here can be used by researchers who wish to measure kinetic, kinematic, and spatiotemporal variables in future research. The lack of observed strong correlations between kinetic, kinematic, and spatiotemporal variables supports the value of simultaneous measures across these domains in running research incorporating biomechanical measures. Finally, the similarity in findings across different sampling frames (30 s, 30 steps, and 1 step) suggests that shorter sampling frames are sufficient for representative data for individual runners. These findings have practical implications for researchers planning to carry out studies that employ IMUs to measure the biomechanics of running. The 4-IMU protocol described in this study can be used to provide a comprehensive analysis of real-world kinematic, kinetic, and spatiotemporal variables without the need for advanced laboratory equipment. Furthermore, the comparison of different sampling strategies can inform researchers’ choices regarding subsequent data analysis. Though this is the first study to measure biomechanical variables from all three categories simultaneously, we have compared our results to previously published studies on one or more biomechanical variable categories. To enable the study results to be comparable, we have performed this comparison under subheadings of the variables of forces (N/g), the variables of time (milliseconds), the variables of steps, and the vertical displacement of the centre of mass (centimetres).

In the variables related to forces, both the PPA and the GRF were higher in our study, almost doubling the values of the comparative studies, both in the least similar studies and in those that were most similar. On the other hand, in the comparison of the SA variable, our study was the one that showed lower values, comparing it with studies with the same configuration (minus the number of steps [11]) or with some difference (placement of the sensors in different positions [25,27]). Regarding the PSD variable, the results of our study were within the average if we compare it with other similar studies, being more similar to the study with the exact IMU placement [11].

The CGT variable was found among the mean values of the compared studies, being one of the highest but not the highest, despite the variability in the characteristics of the other studies. The study showed the most similar values where the IMUs were placed close to the tibiae (talus joint [50]). The ST was lower in our study, although there were differences in the placement of the IMU from the comparative study (instead of tibial and lumbar [51]). The FT in our study was the lowest in the comparative study. However, there was little difference to the most similar study [23] and consequently, the variable DF was the highest, more than double that of the most similar study [23]).

The SR was very similar to the studies compared, even reaching the exact figure in one despite the methodological differences (IMU placed in the lumbar instead of the tibia [35]). The study that showed the most significant difference was the one that was carried out on an athletic outdoor track and with the accelerometer placed in the L5-S1 lumbar zone [52].

The COM variable in our study showed the lowest figure compared to the other studies but with very little difference. Note the differences in speed and slope of +1% in the study by Smith et al. [28], and the selection of 20 steps and the average speed in the Schütte study [35]. The most significant correlations were for variables in the same family. For example, we observed a relationship between the variables MASS, BMI, and GRF, having included this last MASS variable as part of its calculation equation, and it is understood that the greater the mass, the greater the GRF will be [37,50]. Similarly, time variables were correlated with each other. It makes sense that if the running cycle was composed of the flight and ground contact phases [53], and both add up to the stride time when one of them was modified, the others were modified. In the selection of 30 s, the relationship between the flight time and the GCT and ST was positive. This fact was unusual. since if the speed remains constant, the ground contact time must decrease by the same proportion as the flight time increases.

Previous studies found an inverse association between the COM of displacement and PS [53,54]. In the present study, this association was found with the variable SR and only afterselecting 30 steps. Therefore, these results followed the trend of other studies. in which no influence of kinetic forces was found with the COM variable [55,56].

Two trends could be observed in the correlations between running descriptor and kinetic variables. A first trend common to the three measurement interval selections (30”, 30 steps and 1 step) is the positive and strong association of the kinetic variable GRFbw with the running descriptor variable PS. The second trend was the moderate positive association after selecting 30 s and 30 steps, between the kinetic variables MPSDL and PPSDL and the running descriptor variables PS, SR, CTT and ST.

Results were obtained for some variables with large standard deviations. In the case of selecting a single step, this may be because although the presented result was representative of one step, it was calculated using the accelerations or gyroscopes of one right step and one left step. Therefore, an SD will be obtained with values almost as high as the result, such as the FT variable with 96.46 ms (SD ± 49.72) and the PPA with 13.74 g (SD ± 10.4). In the 30-s and 30-step selections, large SDs were also found in the three groups of variables. In the time and force variables, this may be due to a primary reason, such as the heterogeneity of the participants. Regarding the strength variables, the high SD in the GRFbw and SA may be because each runner wore different sports shoes, some more appropriate for overground running, others maximalist, or others for sports in general. In these same variables, and the rest with high SD, such as spatiotemporal variables, it may be due to different biotypes and individuals with different masses running at different speeds, or possibly even running on a treadmill without being accustomed to it.

## 5. Limitations and Perspectives

Some considerations must be taken into account when interpreting these results. Firstly, the test subjects were amateur runners, and the male-to-female ratio was skewed towards males. Secondly, the running pace and the subjects’ shoes were not standardized. Thirdly, the foot strike pattern of the runners was not regulated, which could have impacted the study’s outcome, as per some research [57]. Lastly, it is essential to note that the experiment was conducted on a treadmill, not overground, which could also affect the results [58,59].

Although there are no validation studies with the gold standard, it is possible to use the 4-IMU set-up presented in this protocol to extract spatiotemporal, kinetic and kinematic variables. Our data shows that, with this set-up, the time window used for data analysis (30 s, 30 steps or 1 step) does not matter as there were no significant differences in the results.

Combining the kinetic, kinematic, and spatiotemporal variables in the same study allows us to compare the variables and establish causal relationships. This in turn allows researchers to carry out strategies on specific parameters of the running, to alter others that are related or that influence them, and therefore to be able to modify the desired variables. Likewise, due to these three-band comparisons, new indices derived from certain interrelated variables may be developed to simplify and facilitate future parameterization of running biomechanics.

Future studies are required with more participants and a more homogeneous sample that will make it possible to verify which selection of time or steps produces significant differences in the results obtained.

## 6. Conclusions

In this study, the results of measurements in runners measured with IMUs were presented, which align with the latest studies published by other authors and respond to the previously stated objectives. We concluded that, with a set-up of IMUs, it is possible to parameterize the kinetic, kinematic, and spatiotemporal variables of occasional healthy runners on a treadmill. The study found very strong associations between the same family variables in all selections and moderate associations between kinetic (forces) and kinematic (displacement) variables. The temporal variables were associated with the step rate variable in selecting 30 steps and 30 s of data. There were no significant differences between the biomechanics variables when selecting 30 s, 30 steps, or 1 step. Therefore, the findings of this study support the use of a novel set-up of four IMUs to parameterize the biomechanics of running from occasional runners on the treadmill, being able to extract information from variables from different families, both from one step and several temporal segments. Using a protocol with a configuration of four IMUs has several advantages, including reduced experiment costs and the ability to run it outside a laboratory setting. Additionally, this protocol is easy to replicate, allowing for future research on kinetics and kinematics that can involve runners with varying performance levels and can be conducted in their natural outdoor environments. This opens up a new avenue for studies on running performance analysis.

## Figures and Tables

**Figure 1 sensors-24-02191-f001:**
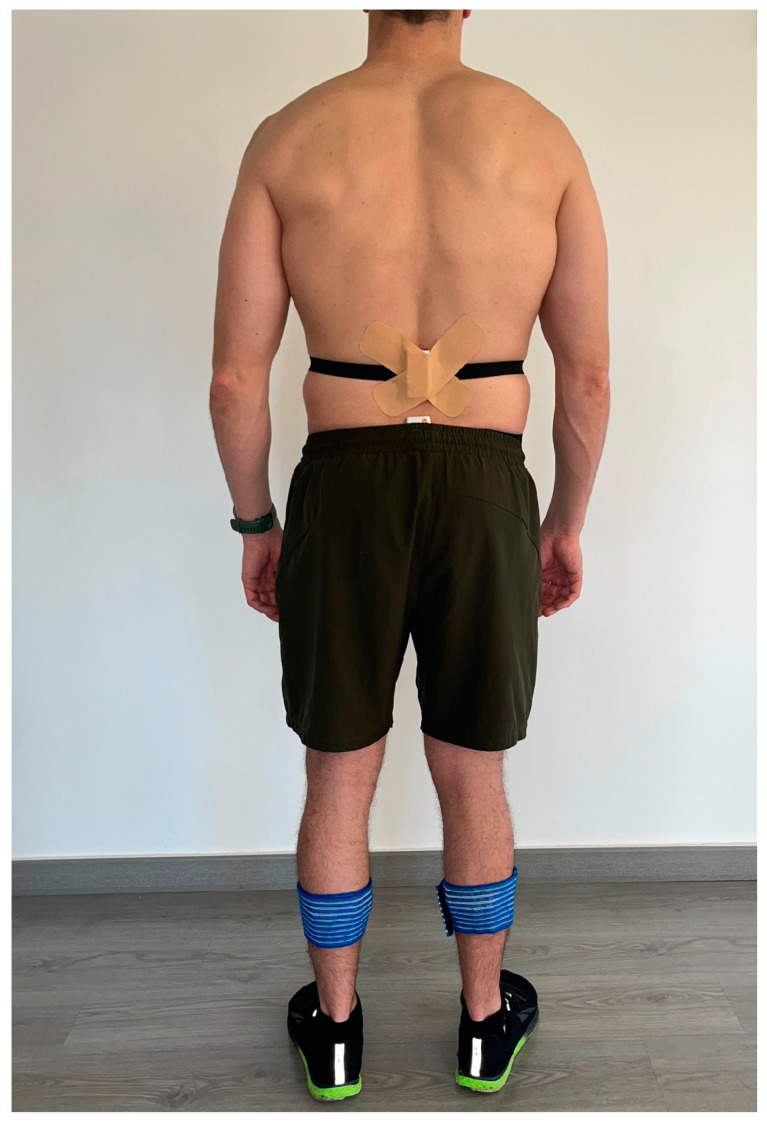
Set-up of 4 IMUs.

**Figure 2 sensors-24-02191-f002:**
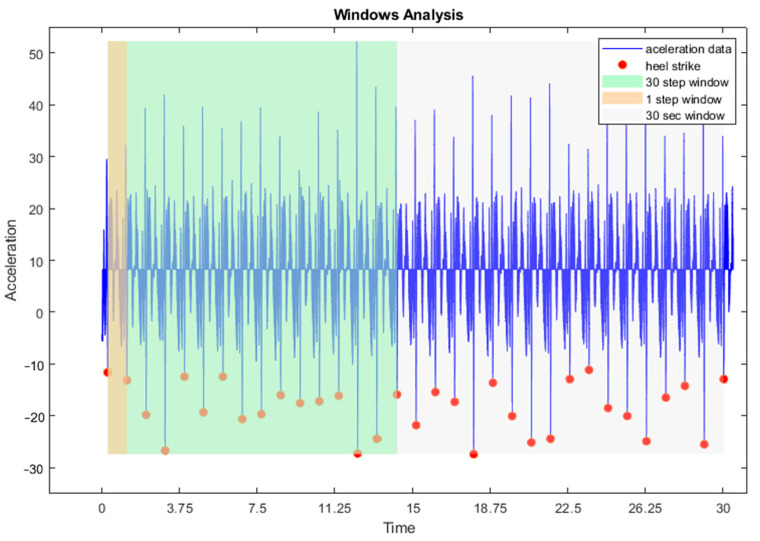
Windows Analysis.

**Table 1 sensors-24-02191-t001:** Demographic characteristics of the participants (*n* = 22), described as mean (±standard deviation).

Characteristics	Male (18)	Female (4)
Age (years)	29 (±6)	26 (±2)
Lower Limb Length (cm)	92 (±6)	86 (±3)
Height (cm)	176 (±7)	159 (±2)
Mass (kg)	77 (±9)	55 (±4)
BMI (kg/m^2^)	0.22 (±0.02)	0.17 (±0.01)

BMI: body mass index.

**Table 2 sensors-24-02191-t002:** Spatiotemporal variables, described as mean and standard deviation (SD).

		Preferred Speed (km/h)	Step Rate (Steps/min)	Stride Time (ms)	Ground Contact Time (ms)	Flight Time (ms)	Duty Factor (%)	N Steps
30 s	Mean	10.1	162.5	376.96	298.2	78.76	0.77	41.2
	SD	1.9	13.4	35.5	31.14	7.29	0.02	2.6
30 Steps	Mean	10.1	156.3	377.93	301.27	76.66	0.78	30
	SD	1.9	13.8	33.65	34.3	7.4	0.02	30
1 Step	Mean	10.1	-	387.83	291.37	96.46	0.76	1
	SD	1.9	-	53.49	21.72	49.72	0.09	1

**Table 3 sensors-24-02191-t003:** Kinetic and kinematic variables. Data presented as mean and standard deviation (SD).

		Ground Reaction Force (N)	GRF Body Weight (BW)	Peak Power T12-L1 (g^2^/Hz)	Peak Power L5-S1 (g^2^/Hz)	Mean Power T12-L1 (g^2^/Hz)	Mean Power L5-S1 (g^2^/Hz)	Mean Shock Attenuation (dB/Hz)	COM (cm)	PPA (g)	ROM (Degrees)
30 s	Mean	3192.17	4.27	0.001	0.0031	0.0004	0.0017	−0.99	9.74	13.12	72.27
	SD	702.9	0.3	0.0056	0.0028	0.0025	0.0015	4.8	1.64	4.42	7.18
30 Steps	Mean	3202.1	4.27	0.0017	0.0046	0.0004	0.0017	−0.81	9.63	13.74	72.3
	SD	704.6	0.3	0.0067	0.0028	0.0025	0.0015	4.87	1.7	5.35	7.18
1 Step	Mean	3287.76	4.26	0.0012	0.0032	0.0004	0.0017	−0.7	9.65	13.74	72.03
	SD	694.5	0.3	0.0059	0.0029	0.0025	0.0015	5.16	1.7	10.4	7.07

GRF: ground reaction force, C0M: centre of mass, PPA: peak positive tibial acceleration, ROM: range of motion.

**Table 4 sensors-24-02191-t004:** Linear regression model variables in 30”, 30 steps and 1 step (*n* = 22).

	Model Summary					Standardized Coefficients		
	Dependent Variable	Explanatory Variables	Weighing	R^2^	SE	Beta	t	*p*
30”	SR ^a^	ST ^c^	BMI	0.952	1.38	−0.976	−19.964	0.000
		FT ^d^		0.990	0.65	−1.831	−9.370	0.000
		DF ^e^	BMI			−1.744	−8.227	0.000
		GCT ^f^				1.182	4.819	0.000
	GCT	MPSDL ^g^	BMI	0.209	0.026	−0.457	−2.296	0.033
		PPSDL ^h^	BMI	0.216	0.026	−0.465	−2.349	0.029
	BMI	DF		0.209	0.023	0.457	2.298	0.032
	ST	MPSDL	BMI	0.229	0.029	−0.479	−2.440	0.024
		PPSDL	BMI	0.238	0.029	−0.488	−2.502	0.021
	GRFbw ^b^	FT		0.293	0.259	−0.541	−2.878	0.009
	ROM ^k^	SA	BMI	0.248	2.418	−0.498	−2.571	0.018
	ROM	PPSDM	BMI	0.218	2.467	−0.467	−2.360	0.029
30 Steps	SR	ST	BMI	0.691	2.43	−8.31	−6.691	0.000
		FT	BMI	0.936	1.16	−2.407	−8.543	0.000
		DF				−3.752	−7.250	0.000
		GCT				1.815	4.832	0.000
		COM ^i^	BMI	0.258	3.77	−0.508	−2.634	0.016
		PPSDL	BMI	0.283	3.7	0.532	2.812	0.011
		MPSDL	BMI	0.307	3.64	0.554	2.979	0.007
	GCT	MPSDL	BMI	0.205	0.55	−0.453	−2.27	0.034
		PPSDL	BMI	0.216	0.55	−0.465	−2.349	0.029
	ST	MPSDL	BMI	0.201	0.029	−0.448	−2.243	0.036
		PPSDL	BMI	0.212	0.029	−0.460	−2.318	0.031
1 Step	GCT	MPSDL	BMI	0.231	0.017	−0.480	−2.449	0.024
		PPSDL	BMI	0.220	0.017	−0.469	−2.374	0.028
	FT	PPA ^j^	BMI	0.260	0.039	0.510	2.654	0.015
	DF	PPA	BMI	0.268	0.036	−0.551	−2.951	0.008
	ROM	PPSDM	BMI	0.203	2.44	−0.450	−2.257	0.035

^a^ SR: Step Rate, ^b^ GRFbw: Ground Reaction Force by Body Weight, ^c^ ST: Stride Time, ^d^ FT: Flight Time, ^e^ DF: Duty Factor, ^f^ GCT: Ground Contact Time, ^g^ MPSDL: Mean Power Spectral Density Low, ^h^ PPSDL: Peak Power Spectral Density Low, ^i^ COM: Vertical Center of Mass displacement, ^j^ PPA: Peak Positive Acceleration of the Tibia, ^k^ ROM: Range of Motion of the Knee.

## Data Availability

The required documentation will be sent to the corresponding author.

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
