# Peer review of "Parameterization of Biomechanical Variables through Inertial Measurement Units (IMUs) in Occasional Healthy Runners"

_sensors, 2024, doi:10.3390/s24072191_

Round 1
Reviewer 1 Report
Comments and Suggestions for Authors
Dear Authors,
Special thanks for your manuscript submission and to the editors for providing the opportunity to review this work.
The study, titled “Parameterization of biomechanical variables through inertial measurement units (IMUs) in occasional healthy runners”. The objectives of that study are to parameterize the biomechanical variables of the running, that is, through a measurement set-up with IMUS to extract the kinematic and spatiotemporal variables and to estimate the kinetic variables, and to describe and establish causal relationships of this biomechanics of the running with descriptive and inferential statistics.
Here are some insights and suggestions:
1. This research may be of interest to Sensors’ readers.
2. It is a suitable topic for fundamental study.
3. Overall, the manuscript sections are well written.
4. Could you write a section that includes the research design instead of lines 84-85?
5. Line 88: The participants were (18-m, 4-f); did the authors find any differences between male and female in the current study or were they not investigated?
6. The procedures and experimental protocol are straightforward and well-written.
7. Line 206: It is preferable to add the description of males and females separately (Tale 1).
8. Line 286: It is preferable to remove Table 5 and use the results of the previous studies to support the introduction and discussion due to the fact that your manuscript does not review.
9. Line 351: Due to the importance of this methodology, it is preferable to introduce a separate section for limitations and future work to clarify.
10. The conclusions section should be improved and include details about the importance of the study results.
11. The references used in this study are relevant and suitable.
12. Finally, this scientific work is both interesting and informative to the reader.
Author Response
Rebuttal Letter
Manuscript ID sensors-2848601
PARAMETERIZATION OF BIOMECHANICAL VARIABLES THROUGH INERTIAL MEASUREMENT UNITS (IMUs) IN OCCASIONAL HEALTHY RUNNERS
Editor:
Comments to the Author:
This research may be of interest to Sensors’ readers.
Authors: The authors want to thank to the Editor for the time invested. The document has been modified according to the comments.
- It is a suitable topic for fundamental study.
Authors: Thank you for the comment.
- Overall, the manuscript sections are well written.
Authors: Thank you for the appreciation.
- Could you write a section that includes the research design instead of lines 84-85?
Authors: Thank you for your suggestion, we created a new section as you suggested:
“2. Materials and Methods
2.1 Design
That is a descriptive cross-section laboratory study. All measurements were carried out on the same day, and there was no follow-up.”
- Line 88: The participants were (18-m, 4-f); did the authors find any differences between male and female in the current study or were they not investigated?
Authors: Thank you for the question. We must admit that they were not investigated because it was not the primary objective. Additionally, the sample size of females needed to be more significant to make any comparison (only four females out of 18 males). In our future research, we will consider this and ensure that we have an equal number of male and female participants.
- The procedures and experimental protocol are straightforward and well-written.
Authors: Thank you for the remark.
- Line 206: It is preferable to add the description of males and females separately (Tale 1).
Authors: Thanks to this suggestion, we have changed Table 1 of demographic characteristics and have presented the data for women and men, also describing it in the text:
“…were selected, of which 18 were male (age: 29 ± 6 years, lower limb length: 92 ± 6 cm, heigh: 176 ± 7 cm, mass: 77 ± 9 kg, body mass index: 0.22 kg/m2 ± 0.02) and 4 were female (age: 26 ± 2 years, lower limb length: 86 ± 3 cm, heigh: 159 ± 2 cm, mass: 55 ± 4 kg, body mass index: 0.17 kg/m2 ± 0.01) (Table 1).”
Table 1. Demographic characteristics of the participant described as Mean ± standard deviation (n=22).
Characteristics |
Male (18) |
Female (4) |
Age (years) |
29 (± 6) |
26 (± 2) |
Lower Limb Length (cm) |
92 (± 6) |
86 (± 3) |
Heigh (cm) |
176 (± 7) |
159 (± 2) |
Mass (kg) |
77 (± 9) |
55 (± 4) |
BMI (kg/m2) |
0.22 (± 0.02) |
0.17 (± 0.01) |
Mean ± SD. BMI: body mass index
- Line 286: It is preferable to remove Table 5 and use the results of the previous studies to support the introduction and discussion due to the fact that your manuscript does not review.
Authors: Thank you for the comment. We eliminated Table 5 and its references in the text and utilized the paper´s references in the discussion and introduction to improve it.
- Line 351: Due to the importance of this methodology, it is preferable to introduce a separate section for limitations and future work to clarify.
Authors: Thank you for your suggestion. We created a new section to develop limitations and future work:
“5. Limitations and perspectives”
- The conclusions section should be improved and include details about the importance of the study results.
Authors: Thank you for the comment. We changed the conclusions paragraph.
- The references used in this study are relevant and suitable.
Authors: Thank you for the remark.
- Finally, this scientific work is both interesting and informative to the reader.
Authors: Thank you very much again for your appreciation.

Reviewer 2 Report
Comments and Suggestions for Authors
The manuscript presented results of various running related measures, using correlation and multivariate regression analyses. While the results seem to be interesting and add value to the literature, there are some major concerns.
- I don't think based on the way regression analysis was used, the authors cannot say predictors.
- English. I believe there are many errors and mistakes in English expression and mechanics. I strongly suggest the authors to seek professional proof reading services.
Specific comments
- Abstract: "were not correlations of kinetic variables (forces) with the kinematic (displacement)"
- Page 1, Line 28: "Within these internal factors..." It is unclear what "these" refer to
- Page 1, Lines 29-31: I am not sure what the purpose of "we find biomechanical factors ..."
- Page 2, Lines 64-81: I think this part is a bit strange. A new paragraph can start with "this study aims to add value ...". The paragraph (Page 2, Line 70-76) seems to be more appropriate for the discussion or conclusion.
- Page 2, Line 80: I don't think the current study was designed to establish causal relationships, though can show correlations among different measures.
- Page 2, Line 86: Is there a reason for"<" three runs per week? I feel that 2-3 times can be viewed as occasional.
- Page 2, Line 87-88: "... students of the faculty" I don't know who is "the" faculty.
- Page 3, Line 118: was the treadmill instrumented? If so, please state that; otherwise it is unclear how GRFs were obtained. The model and manufacturer information is given for the treadmill here, yet the corresponding information for IMUs were given below. Can you make it consistent?
- Page 3, Line 119: It is unclear why data were used only from the 6th minute. Could you elaborate?
- Page 3, Line 124: what instruction was used for the preferred speed? preferred speed for easy run, long run, 10k, etc.?
- Page 3, Line 126: collides -> contacts
- Page 4, Line 137: It is unclear what "g2" means
- Page 4, Lines 150: Please elaborate how three windows were determined. Can you include a figure to visually shows these three windows? Does the 1st windows include only support phases? assume that these windows were from the 6th minutes, correct? If so, please clarify.
- Page 4, Line 152-154: The sentence is difficult to follow. I suggest rephrasing.
- Page 4, Line 153-155: Please provide relevant citations used to extract the noted measures.
- Page 4, Line 156: Please use a different verb other than "carried out".
- Page 4, Section 2.4: Please state that no filtering was performed, given no information was provided about filtering.
- Page 4, Line 164: Is it the whole-body COM? If so, please state so.
- Page 4, Line 167: "less in the selection of" I am not sure what this means.
- Page 5, Line 182: statics? Would it be a typo?
- Page 5, Line 187: I think in the current form, I would say explanatory variables, since the authors did not actually example new observations using predictor(s). That is, validation is missing. And did you consider including any covariates (e.g., preferred speed) or potentially important anthropometric measures (e.g., height, body mass, etc.)? If not, please explain why. (In the results, I saw that BMI was adjusted, though not noted in the statistical analysis sections)
- Page 5, Line 188: Would a one-way ANOVA have been more efficient? If fact, it could have been two factors, if you considered the category of measures as a factor.
- Page 6, Table 1: with BMI, it is unclear what "m-2" means.
- Page 6, Line 221: A one-step analysis was described in the statistical analysis section.
- Page 6, Lines 221-228: It seems this part is just a repeat of the information in Tables 2 and 3. Not sure if this paragraph is useful.
- Page 6, Line 223: "this right-selected" It's unclear what this means.
- Page 6, Tables 2 and 3: can you report statistical differences between the windows (i.e., 30 sec, 30 steps, 1 step)?
- Page 6, Table 2: It is unclear what the superscript 0 means in N Steps.
- Page 7, Line 239: Could you at least give some idea about the direction of correlations? generally positive, negative, or mixed?
- Page 7, Line 244: "In selecting 30 seconds and 1 step," Not sure what this means. Why do you need to select 30 seconds and 1 step now?
- Page 11, Line 325: does "inverse connection" mean a negative correlation?
- Page 11, Line 330: "No significant linear regressions" this doesn't make sense
- Page 11, Line 336: "moderate regressions" I am not sure what this means.
Comments on the Quality of English Language
As noted in my review, the current manuscript requires a substantial editing.
Author Response
REVIEWER 2
Comments and Suggestions for Authors
The manuscript presented results of various running related measures, using correlation and multivariate regression analyses. While the results seem to be interesting and add value to the literature, there are some major concerns.
- I don't think based on the way regression analysis was used, the authors cannot say predictors.
- English. I believe there are many errors and mistakes in English expression and mechanics. I strongly suggest the authors to seek professional proof reading services.
Specific comments
- Abstract: "were not correlations of kinetic variables (forces) with the kinematic (displacement)"
AUTHORS: Thank you for the comment. It has been corrected.
- Page 1, Line 28: "Within these internal factors..." It is unclear what "these" refer to
AUTHORS: Thank you for the remark. It has been corrected.
- Page 1, Lines 29-31: I am not sure what the purpose of "we find biomechanical factors ..."
AUTHORS: Thank you for your comment. We have made a change: “Intrinsic factors include biomechanical variables, categorised as kinetic, kinematic, spatiotemporal or running descriptors, and neuromuscular factors.”
- Page 2, Lines 64-81: I think this part is a bit strange. A new paragraph can start with "this study aims to add value ...". The paragraph (Page 2, Line 70-76) seems to be more appropriate for the discussion or conclusion.
AUTHORS: Thank you for your appreciation. We made the changes he suggested: we started a new paragraph with "this study aims to add value..." and moved the paragraph from lines 70-76 to the discussion section.
- Page 2, Line 80: I don't think the current study was designed to establish causal relationships, though can show correlations among different measures.
AUTHORS: Thank you for the remark. We have changed that line since the original expression was incorrect:
“…to describe the relationships between these variables, and to investigate the effect of different data sampling approaches.”
- Page 2, Line 86: Is there a reason for"<" three runs per week? I feel that 2-3 times can be viewed as occasional.
AUTHORS: Thank you for expressing your appreciation. As mentioned, running 2-3 times per week is a feasible frequency for occasional runners. However, we would like to emphasize that this frequency should not exceed three times per week. We have revised the text to ensure that this information is more precise.
“occasional runners (< 50 km per week or 2-3 three runs per week, and no less than 5 km per week) were selected”.
- Page 2, Line 87-88: "... students of the faculty" I don't know who is "the" faculty.
AUTHORS: Thank you for your comment. We have corrected the grammatical mistake:
“They were students from the Faculty and participated voluntarily”
- Page 3, Line 118: was the treadmill instrumented? If so, please state that; otherwise it is unclear how GRFs were obtained. The model and manufacturer information is given for the treadmill here, yet the corresponding information for IMUs were given below. Can you make it consistent?
AUTHORS: Thank you for your appreciation. The treadmill was not instrumented, and we have specified it this way:
“The measurement test was a run of 6 minutes on a noninstrumented treadmill…”
The model and manufacturer information of the IMUs has been changed to the same paragraph as the treadmill information (2.3 Experimental Procedure and Data Collection).
The GRF variable was calculated by estimating the accelerations coming from the IMUs; it was not a direct measurement:
“We used the accelerometry data from the IMUs to estimate the kinetic variables” (line 257), section 2.5.3 Kinetic variables.
- Page 3, Line 119: It is unclear why data were used only from the 6th minute. Could you elaborate?
AUTHORS: Thank you for the comment. It has been corrected:
”After accommodating to the treadmill for the first 5 minutes [39], the sixth minute of running was recorded. The three data sections were selected from that minute: the first 30 seconds, the first 30 steps, and the first step.”
- Page 3, Line 124: what instruction was used for the preferred speed? preferred speed for easy run, long run, 10k, etc.?
AUTHORS: Thank you for the remark. We have specified it in the section 2.3 Experimental Procedure and Data Collection:
“Participants were instructed to run at a self-selected speed equivalent to the speed they would normally employ for a 10k run.”
- Page 3, Line 126: collides -> contacts
AUTHORS: Thank you for the comment. It has been corrected.
- Page 4, Line 137: It is unclear what "g2" means
AUTHORS: Thank you for the remark. It has been corrected. We wanted to say (g2/Hz)
- Page 4, Lines 150: Please elaborate how three windows were determined. Can you include a figure to visually shows these three windows? Does the 1st windows include only support phases? assume that these windows were from the 6th minutes, correct? If so, please clarify.
AUTHORS: Thank you for your appreciation. The time windows were selected from 6 minutes after 5 minutes of accommodation on the treadmill. The correctness of the explanation of the selection of the windows has been modified:
“(1) in the first window, completed steps were collected over 30 seconds.; (2) In the second, the first 30 complete steps, 15 with each leg, were recorded; and (3) the third window, in which the first complete step of each foot was analysed”.
In addition to changing the explanation, we added the visual figure (Figure 2) representing the three windows.
- Page 4, Line 152-154: The sentence is difficult to follow. I suggest rephrasing.
AUTHORS: Thank you for the remark. We have rewritten those lines to make them easier to understand:
“Once the time windows and steps were selected, the specific variables were calculated as part of the process. The formulas utilized either accelerometry data, gyroscope da-ta, or a combination of both, depending on the specific variable being calculated. Additionally, one or more IMUs were used to measure various parts of the body, including the left or right tibia, both tibias, sacrum S1, lumbar L1, or a combination of sacrum and lumbar, depending on the variable being calculated.”
- Page 4, Line 153-155: Please provide relevant citations used to extract the noted measures.
AUTHORS: We appreciate your comment. We have changed the text to make it easier to understand:
“The formulas utilized either accelerometry data, gyroscope data, or a combination of both, depending on the specific variable being calculated. Additionally, one or more IMUs were used to measure various parts of the body, including the left or right tibia, both tibias, sacrum S1, lumbar L1, or a combination of sacrum and lumbar, depending on the variable being calculated.”
In order to calculate the variables of this study, we have used different articles that employ different data extraction formulas. We have cited the relevant formula for each variable in the following sections: 2.5.1 Running descriptors, 2.5.2 Kinematic variables, and 2.5.3 Kinetic variables.
- Page 4, Line 156: Please use a different verb other than "carried out".
AUTHORS: Thank you for the comment. It has been corrected:
“In order to make the process clearer, a summary table has been created:”
- Page 4, Section 2.4: Please state that no filtering was performed, given no information was provided about filtering.
AUTHORS: Thank you for your appreciation. Because we have used several types of filters depending on which variable we wanted to obtain and using what data (accelerometry, gyroscope, or both), we have provided the filter information in the following sections where each variable is explained: 2.5.1 Running descriptors 2.5.2 Kinematic variables, and 2.5.3 Kinetic variables.
- Page 4, Line 164: Is it the whole-body COM? If so, please state so.
AUTHORS: Thank you for the comment. Yes, it is he whole-body COM. It has been corrected.
- Page 4, Line 167: "less in the selection of" I am not sure what this means.
AUTHORS: Thank you for the remark. We have rewritten those lines to make them easier to understand:
“We used the accelerometry data from the IMUs to estimate the kinetic variables. We calculated the Root Mean Square (RMS) and the ground reaction force (GRF) by taking the means of the data in 30-second and 30-step intervals.”
- Page 5, Line 182: statics? Would it be a typo?
AUTHORS: Thank you for the remark. It has been corrected.
- Page 5, Line 187: I think in the current form, I would say explanatory variables, since the authors did not actually example new observations using predictor(s). That is, validation is missing. And did you consider including any covariates (e.g., preferred speed) or potentially important anthropometric measures (e.g., height, body mass, etc.)? If not, please explain why. (In the results, I saw that BMI was adjusted, though not noted in the statistical analysis sections)
AUTHORS: Thank you for the comment. We modified the line and the paragraph, and it now looks like this:
“A one-factor ANOVA analysis was performed, and subsequently, a Bonferroni Posthoc test was performed to specify the differences between variables in the three different groups of 30 seconds, 30 steps and 1 step.”
We specified the point that the linear regression was adjusted by BMI:
“In addition, multivariate linear regression models were performed and adjusted by BMI.”
- Page 5, Line 188: Would a one-way ANOVA have been more efficient? If fact, it could have been two factors, if you considered the category of measures as a factor.
AUTHORS: Thank you for the remark. It is possible that a one-way ANOVA would be more efficient. However, we did it this way because we were missing the graphs of some variables at the same time we calculated it
- Page 6, Table 1: with BMI, it is unclear what "m-2" means.
AUTHORS: Thank you for the remark. It has been corrected.
- Page 6, Line 221: A one-step analysis was described in the statistical analysis section.
AUTHORS: Thank you for the comment. The paragraph of that line has been eliminated as you have suggested in the following comment.
- Page 6, Lines 221-228: It seems this part is just a repeat of the information in Tables 2 and 3. Not sure if this paragraph is useful.
AUTHORS: Thank you for the remark. The paragraph has been deleted.
- Page 6, Line 223: "this right-selected" It's unclear what this means.
AUTHORS: Thank you for the comment. The paragraph of that line has been eliminated as you have suggested in the previous comment.
- Page 6, Tables 2 and 3: can you report statistical differences between the windows (i.e., 30 sec, 30 steps, 1 step)?
AUTHORS: Dear reviewer, we wanted to inform you that we have relocated the sentence in which we mention the statistical differences among the three windows (Bonferroni test). According to this test, there were no significant differences between any of the variables, including the new Knee ROM variable, in any of the three groups. If we have misunderstood your comment, please let us know so that we can make the necessary changes.
- Page 6, Table 2: It is unclear what the superscript 0 means in N Steps.
AUTHORS: Thank you for the remark. It has been corrected.
- Page 7, Line 239: Could you at least give some idea about the direction of correlations? generally positive, negative, or mixed?
AUTHORS: Thank you for the comment. "We have removed the paragraph in question and substituted it with the one on line 361:
“In the three correlation arrays, very strong, strong and moderate associations were found, being mixed since the direction was positive or negative depending on the variable type.”
- Page 7, Line 244: "In selecting 30 seconds and 1 step," Not sure what this means. Why do you need to select 30 seconds and 1 step now?
AUTHORS: Thank you for your comment. We have edited the paragraph:
“Correlations between variables were calculated for the different sampling strategies of 30 seconds, 30 steps and 1 step (supplementary data 3). Very strong associations were only between the variables whose formulas share the same unit of measurement.”
- Page 11, Line 325: does "inverse connection" mean a negative correlation?
AUTHORS: Thank you for the remark. It has been corrected:
“In previous studies a negative association was found between…”
- Page 11, Line 330: "No significant linear regressions" this doesn't make sense
AUTHORS: Thank you for your comment. We have removed the paragraph containing that line and replaced it with a better-worded version.
- Page 11, Line 336: "moderate regressions" I am not sure what this means.
AUTHORS: Thank you for your comment. We have removed the paragraph containing that line and replaced it with a better-worded version.

Reviewer 3 Report
Comments and Suggestions for Authors
I: General Evaluation
The study employed four IMUs fixed on the medial- anterior aspect of the shin and skin overlying the T12/L1 and L5/S1 vertebrae levels (mid-back, low-back) to collect and quantify biomechanical variables during treadmill running. The study aimed to investigate the differences between data collected under different running conditions. However, the significance and practical contribution of the study are not clearly reflected.
II: Specific Recommendations
(A) Title and Abstract Section
1. The title implies that occasional health runners were selected, but it doesn't explain why this population was chosen, potentially affecting the applicability of the results and findings.
2. The abstract didn't explicitly mention that the participants were occasional health runners.
3. The abstract failed to briefly mention the study's applications and practical contributions, only describing the results shown by experimental data.
(B) Introduction Section
1. The rationale for determining one step, 30 steps, and 30 seconds in the running cycle is not adequately explained.
2. The introduction could provide a brief summary of previous major studies, leading to the innovative points and significance of the current research.
(C) Materials and Methods Section
1. The study involving the collection of shock attenuation data from the lumbar region should exclude participants with lumbar diseases or deformities.
2. Knee joint Range of Motion (ROM) data could be included in the kinematic variables, as gyroscope-equipped IMUs may be convenient and feasible to collect this data.
3. Figure 1 (setup of four IMUs) is better presented with real images, as readers cannot see how the fixing method using rubber bands and elastic bands might affect the force exerted or joint angles.
(D) Results Section
Data obtained from the 30s running include average stride time of 376.96 ms (SD ±35.5) and ground contact time of 298.2 ms (SD ±31.14); 30 steps running data show average stride time of 377.93 ms (SD ±33.65) and average ground reaction force of 3202.1 N (SD ±704.6); data from the one-step experiment include flight time of 96.46 ms (SD ±49.72) and stride time of 387.83 ms (SD ±53.49). The large standard deviations in these data points should be explained and analyzed.
(E) Discussion Section
1. This section failed provide further explanations and analyses of the experimental results, nor adequately explain the use of IMUs to estimate biomechanical variables, as mentioned in the Introduction.
2. Limitations of the experimental design and methods, as well as potential areas for improvement, are not well-discussed. The study does not provide detailed explanations for the significant deviations observed in certain data points.
3. The discussion lacks specific details on how the research results could be practically applied and in which field.
(F) Conclusion
Overall, the significance of current study appears to be vague, and despite a wealth of experimental data, it failed to offer valuable insights and information. The data in the Table of this article is not properly arranged, and the font is blunt and bold. The font inconsistency between English letters and numbers in the same table failed to facilitate reading and understanding.
Comments on the Quality of English LanguageThe writing is fine, but polish from native English speaker is suggested to improve the clarity.
Author Response
Rebuttal Letter
Manuscript ID sensors-2848601
PARAMETERIZATION OF BIOMECHANICAL VARIABLES THROUGH INERTIAL MEASUREMENT UNITS (IMUs) IN OCCASIONAL HEALTHY RUNNERS
REVIEWER 3
I: General Evaluation
The study employed four IMUs fixed on the medial- anterior aspect of the shin and skin overlying the T12/L1 and L5/S1 vertebrae levels (mid-back, low-back) to collect and quantify biomechanical variables during treadmill running. The study aimed to investigate the differences between data collected under different running conditions. However, the significance and practical contribution of the study are not clearly reflected.
II: Specific Recommendations
(A) Title and Abstract Section
- The title implies that occasional health runners were selected, but it doesn't explain why this population was chosen, potentially affecting the applicability of the results and findings.
AUTHOR: Thank you for your suggestion. We created a new section as you suggested in section 2. 2 Participants:
“In order for the sample to be as representative as possible of the general population, Twenty-two healthy occasional runners (running distance between 5-50k/week; 2-3 runs/week) were selected.”
- The abstract didn't explicitly mention that the participants were occasional health runners.
AUTHOR: Thank you for the appreciation. We have modified it.
- The abstract failed to briefly mention the study's applications and practical contributions, only describing the results shown by experimental data.
We have modified the final 2 sentences of the Abstract to include reference to practical implications:
“Our results suggest that a 4-IMU setup, as presented in this study, is a viable approach for parameterization of the biomechanical variables in running and there are no significant differences in the biomechanical variables studied independently if we select data from 1 step, 30 steps or 30 seconds for processing and analysis. These results can assist in protocol design in future running research”
(B) Introduction Section
- The rationale for determining one step, 30 steps, and 30 seconds in the running cycle is not adequately explained.
AUTHOR: Thank you for your suggestion. There is no pre-established measurement protocol for measuring the time or steps needed to measure different variables. The first step is to determine if measuring the minimum possible is sufficient in providing the required information compared to other variables. The thirty seconds and steps were to determine a specific and comparable amount to compare the variables meaningfully. We created a new paragraph as you suggested:
“It is essential to consider the time or steps taken to determine the duration or distance of measurement protocols. Sometimes, different studies with different samples yield similar results. For instance, in a study by Smith et al [28] the COM variable was measured in runners for 30 seconds, while in a study by Schütte et al. [35] 20 steps were measured, resulting in almost the same displacement in centimetres. However, studies usually yield different results if they do not have the same sample period, making it difficult to compare them. This sample disparity is evident in the review by Manson et al. [8], where out of 131 articles, 50 selected a certain number of steps or strides, while 42 selected a period equal to or less than 60 seconds.”
- The introduction could provide a brief summary of previous major studies, leading to the innovative points and significance of the current research.
AUTHORS: Thank you for your appreciation. We have added new paragraphs to emphasize the current research.
(C) Materials and Methods Section
- The study involving the collection of shock attenuation data from the lumbar region should exclude participants with lumbar diseases or deformities.
AUTHORS: Thank you for the remark. We have included that aspect as you required:
“A participant who presented (i) concurrent lower limb injuries and, (ii) lumbar dis-eases or deformities, and (iii) the presence of rheumatoid, neurological, or degenerative diseases was excluded.”
- Knee joint Range of Motion (ROM) could be included in the kinematic variables, as gyroscope-equipped IMUs may be convenient and feasible to collect this data.
AUTHORS: Thank you for the comment. We have included this variable in the study.
- Figure 1 (setup of four IMUs) is better presented with real images, as readers cannot see how the fixing method using rubber bands and elastic bands might affect the force exerted or joint angles.
AUTHORS: Thank you for your suggestion. We have changed the Figure 1 for a real picture of the set-up.
(D) Results Section
Data obtained from the 30s running include average stride time of 376.96 ms (SD ±35.5) and ground contact time of 298.2 ms (SD ±31.14); 30 steps running data show average stride time of 377.93 ms (SD ±33.65) and average ground reaction force of 3202.1 N (SD ±704.6); data from the one-step experiment include flight time of 96.46 ms (SD ±49.72) and stride time of 387.83 ms (SD ±53.49). The large standard deviations in these data points should be explained and analyzed.
AUTHORS: Thank you for the comment. We responded to the identified SDs by adding a new paragraph to the DISCUSSION section:
“Results were obtained for some variables with large standard deviations. In the case of selecting a single step, this may be because although the presented result was representative of one step, it was calculated using the accelerations or gyroscopes of one right step and one left step. Therefore, an SD will be obtained with values almost as high as the result, such as the FT variable with 96.46 ms (SD ±49.72) and the PPA with 13.74 g (SD ±10.4). In the 30-second and 30-step selections, large SDs were also found in the three groups of variables. In the time and force variables, this may be due to a primary reason, such as the heterogeneity of the participants. Regarding the strength variables, the high SD in the GRFBW and SA may be because each runner wore different sports shoes, some more appropriate for overground running, others maximalist, or others for sports in general. In these same variables, and the rest with high SD, such as spatiotemporal variables, it may be due to different biotypes and individuals with different masses running at different speeds, even adding to this the fact of running on a treadmill without being accustomed to it.”
(E) Discussion Section
- This section failed provide further explanations and analyses of the experimental results, nor adequately explain the use of IMUs to estimate biomechanical variables, as mentioned in the Introduction.
We have added a paragraph at the beginning of the discussion section to provide an interpretation of the observed results, as well as their potential implications for future research in this field.
Our results suggest that the 4-IMU approach to data capture, along with the data processing strategies described in the methods section, represents a feasible approach to simultaneously measuring kinetic, kinematic and spatiotemporal variables during treadmill running. The data capture process was easy to implement and the method presented here can be used by researchers who wish to measure kinetic, kinematic and spatiotemporal variables in future research. The lack of observed strong correlations between kinetic, kinematic and spatiotemporal variables provides supports the value of simultaneous measures across these domains in running research that incorporates biomechanical measures. Finally, the similarity in findings across different sampling frames (1 step, 30 steps, 30 seconds) suggests that shorter sampling frames are sufficient for representative data for individual runners.
- Limitations of the experimental design and methods, as well as potential areas for improvement, are not well-discussed. The study does not provide detailed explanations for the significant deviations observed in certain data points.
AUTHORS: Thank you for your suggestion, we created a new section:
- Limitations and perspectives
- The discussion lacks specific details on how the research results could be practically applied and in which field.
This has been addressed in the response to the first point above.
(F) Conclusion
Overall, the significance of current study appears to be vague, and despite a wealth of experimental data, it failed to offer valuable insights and information. The data in the Table of this article is not properly arranged, and the font is blunt and bold. The font inconsistency between English letters and numbers in the same table failed to facilitate reading and understanding.
AUTHORS: Thank you for the appreciation. We have modified the tables by unifying the source to improve visualization and emphasized the importance of the current study.
Comments on the Quality of English Language
The writing is fine, but polish from native English speaker is suggested to improve the clarity.
AUTHORS: The authors want to thank to the Editor for the time invested. The document has been modified according to the comments.

Round 2
Reviewer 2 Report
Comments and Suggestions for Authors
I appreciate the authors for the revised version. Overall, the revised version looks good, though I have some more suggestions.
- Page 1, Line 17: "22 healthy ..." -> "A total of 22 healthy ..."
- Page 1, Line 18: "22 healthy... were measured..." It is unclear what was measured.
- Page 1, Line 20: "... in over selections..." I think this is not grammatically correct
- Page 1, Line 20: "thirty seconds" -> 30 seconds only because you provide Arabic numerals for other numbers (e.g., 30 steps). And should it be in the following order? 1 step, 30 steps, 30 seconds to be consistent with what's shown in Line 27.
- Page 1, abstract: when noting associations, it would be great if the authors reported their directions as well.
- Page 2, Line 58: "Without these established protocols," Does this mean "given the lack of established protocols"?
- Page 2, Line 65: running -> running dynamics
- Page 2, Line 72: "Many studies ..." It seems citations are missing.
- Page 2, Lines 86-88: I feel this paragraph can be part of the previous paragraph.
- Page 2, Line 97: I am not sure what "that study" refers to. And this sentence seems a repeat of the first sentence in this paragraph.
- Page 3, Line 102: Design -> Experimental design
- Page 3, Line 106: I suggest removing this sentence (I don't think that makes much sense), and simply noting a convenience sampling was used.
- Page 3, Lines 107-108: "running distance between 5-50k/week; 2-3 runs/week" should be placed as part of inclusion criteria below.
- Page 3, Table 1: In the table caption, it is noted that "Mean ± standard deviation". Yet, the entry in the table is mean (± standard deviation). Please match the style.
- Page 10, Line 354: "To align our results previous research," it is a bit unclear. Does this mean, to enable the study results comparable,
- Table 4: Predictive variables -> explanatory variables? As noted in the previous review, I don't think the current study is designed to predict.
Comments on the Quality of English LanguageI strongly suggest that the authors consider utilizing professional English editing services.
Author Response
Rebuttal Letter
Manuscript ID sensors-2848601
PARAMETERIZATION OF BIOMECHANICAL VARIABLES THROUGH INERTIAL MEASUREMENT UNITS (IMUs) IN OCCASIONAL HEALTHY RUNNERS
Comments and Suggestions for Authors:
I appreciate the authors for the revised version. Overall, the revised version looks good, though I have some more suggestions.
- Page 1, Line 17: "22 healthy ..." -> "A total of 22 healthy ..."
Authors: Thank you for the comment. It has been corrected.
- Page 1, Line 18: "22 healthy... were measured..." It is unclear what was measured.
Authors: Thank you for your suggestion. It has been corrected.
- Page 1, Line 20: "... in over selections..." I think this is not grammatically correct
Authors: Thank you for the remark. It has been corrected.
- Page 1, Line 20: "thirty seconds" -> 30 seconds only because you provide Arabic numerals for other numbers (e.g., 30 steps). And should it be in the following order? 1 step, 30 steps, 30 seconds to be consistent with what's shown in Line 27.
Authors: Thank you for expressing your appreciation. It has been corrected.
- Page 1, abstract: when noting associations, it would be great if the authors reported their directions as well.
Authors: Thank you for your suggestion. We have provided a detailed explanation of the relationship between groups of variables belonging to the same family. However, regarding the correlation between kinetic and kinematic variables, we have observed positive and negative (inverse) results across different variables and temporal selections. Therefore, we have presented this information in the results section to avoid overcomplicating the summary.
- Page 2, Line 58: "Without these established protocols," Does this mean "given the lack of established protocols"?
Authors: Thank you for the remark. It has been corrected.
- Page 2, Line 65: running -> running dynamics
Authors: Thank you for the comment. It has been corrected.
- Page 2, Line 72: "Many studies ..." It seems citations are missing.
Authors: Thank you for your appreciation. It has been corrected.
- Page 2, Lines 86-88: I feel this paragraph can be part of the previous paragraph.
Authors: Thank you for the comment. It has been corrected.
- Page 2, Line 97: I am not sure what "that study" refers to. And this sentence seems a repeat of the first sentence in this paragraph.
Authors: Thank you for your contribution. It has been corrected.
- Page 3, Line 102: Design -> Experimental design
Authors: Thank you for the comment. It has been corrected.
- Page 3, Line 106: I suggest removing this sentence (I don't think that makes much sense), and simply noting a convenience sampling was used.
Authors: Thank you for your appreciation. It has been corrected.
- Page 3, Lines 107-108: "running distance between 5-50k/week; 2-3 runs/week" should be placed as part of inclusion criteria below.
Authors: Thank you for your suggestion. It has been corrected.
- Page 3, Table 1: In the table caption, it is noted that "Mean ± standard deviation". Yet, the entry in the table is mean (± standard deviation). Please match the style.
Authors: Thank you for your appreciation. It has been corrected.
- Page 10, Line 354: "To align our results previous research," it is a bit unclear. Does this mean, to enable the study results comparable,
Authors: It has been corrected.
- Table 4: Predictive variables -> explanatory variables? As noted in the previous review, I don't think the current study is designed to predict.
Authors: Thank you for the comment. It has been corrected.
Comments on the Quality of English Language:
I strongly suggest that the authors consider utilizing professional English editing services.
Authors: Thank you for your suggestion; we have done a grammar check with a native expert to solve this problem.
